Multi-level machine learning prediction of protein–protein interactions in Saccharomyces cerevisiae

Zubek Julian 1 2
Tatjewski Marcin 1 2
Boniecki Adam 3
Mnich Maciej 4
Basu Subhadip 5
Plewczynski Dariusz 1 dariuszplewczynski@gmail.com
1 Centre of New Technologies, University of Warsaw , Warsaw , Poland
2 Institute of Computer Science, Polish Academy of Sciences , Warsaw , Poland
3 Faculty of Mathematics, Informatics and Mechanics, University of Warsaw , Warsaw , Poland
4 Faculty of Mathematics and Computer Science, Jagiellonian University , Cracow , Poland
5 Department of Computer Science and Engineering, Jadavpur University , Kolkata, West Bengal , India
Wilke Claus
Electronic publication date: 2015 Jul 2
Publication date: 2015
Volume: 3
Electronic Location ID: e1041
Received 2015 Mar 27; Accepted 2015 May 31
Copyright: © 2015 Zubek et al.
Copyright year: 2015
Copyright holder: Zubek et al.
License: This is an open access article distributed under the terms of the Creative Commons Attribution License, which permits unrestricted use, distribution, reproduction and adaptation in any medium and for any purpose provided that it is properly attributed. For attribution, the original author(s), title, publication source (PeerJ) and either DOI or URL of the article must be cited.
License URL: https://creativecommons.org/licenses/by/4.0/

Keywords: Protein-protein interactions, Protein interaction networks, Multi-scale models, Protein sequence, Machine learning, Physico-chemical indices, Interaction patches, Sequence segments, Local sequence-structure segments

Funding: European Social Fund Polish National Science Centre FASTTRACK grant SR/FTP/ETA-04/2012 This article is funded by the European Union from financial resources of the European Social Fund, Project PO KL “Information technologies: Research and their interdisciplinary applications”; 2014/15/B/ST6/05082 and 2013/09/B/NZ2/00121 grants from the Polish National Science Centre; COST BM1405 and BM1408 EU actions. Subhadip Basu’s research was partially supported by the FASTTRACK grant (SR/FTP/ETA-04/2012) by DST, Government of India. The funders had no role in study design, data collection and analysis, decision to publish, or preparation of the manuscript.

==============================
Accurate identification of protein–protein interactions (PPI) is the key step in understanding proteins’ biological functions, which are typically context-dependent. Many existing PPI predictors rely on aggregated features from protein sequences, however only a few methods exploit local information about specific residue contacts. In this work we present a two-stage machine learning approach for prediction of protein–protein interactions. We start with the carefully filtered data on protein complexes available for Saccharomyces cerevisiae in the Protein Data Bank (PDB) database. First, we build linear descriptions of interacting and non-interacting sequence segment pairs based on their inter-residue distances. Secondly, we train machine learning classifiers to predict binary segment interactions for any two short sequence fragments. The final prediction of the protein–protein interaction is done using the 2D matrix representation of all-against-all possible interacting sequence segments of both analysed proteins. The level-I predictor achieves 0.88 AUC for micro-scale, i.e., residue-level prediction. The level-II predictor improves the results further by a more complex learning paradigm. We perform 30-fold macro-scale, i.e., protein-level cross-validation experiment. The level-II predictor using PSIPRED-predicted secondary structure reaches 0.70 precision, 0.68 recall, and 0.70 AUC, whereas other popular methods provide results below 0.6 threshold (recall, precision, AUC). Our results demonstrate that multi-scale sequence features aggregation procedure is able to improve the machine learning results by more than 10% as compared to other sequence representations. Prepared datasets and source code for our experimental pipeline are freely available for download from: http://zubekj.github.io/mlppi/ (open source Python implementation, OS independent).

Introduction

Systems biology and bioinformatics study interactions between various biocomponents of living cells that spans across multiple spatial and temporal scales. The goal is to understand how the complex phenomena arise given the properties of building blocks. Specifically, proteins are characterised in multiple scales: first in the microscale, by their local post-translational modifications; second, by the interactions with metabolites and small chemical molecules (inhibitors); third in the mesoscale, by the three-dimensional structure of active sites, or interaction interfaces; fourth in the macroscale, by the global 3D structure that comprises the macromolecular complexes; and finally in the time-scale, by their dynamical properties related to the changes of their structure, or physico-chemical properties upon participating in the given biophysical process. Such variety of scales, each linked with different biological function is rooted in their complex and spatio-temporal network of interactions with other smaller biomolecules (metabolites, ligands), comparable size proteins, RNA molecules, and much larger DNA macrochains. Starting from reliable information on single, binary interactions it is possible to reconstruct the whole interaction network, therefore providing further insight into proteins’ biological functions on the whole-cell level.

In this paper, we focus on protein–protein interactions. We develop ensemble learning method for identification of proteins binary interactions by predicting residue-residue interactions. Moreover, we demonstrate how to integrate sequence information from the lower scales into the higher scale machine learning predictor. In our approach, binary interaction between two proteins is predicted by considering all possible interactions between their sequence segments using level-I predictor. An output of this phase is the matrix of scores with the size corresponding to the proteins’ lengths. Given the threshold on the likelihoods, this represent the whole network of all possible residue contacts that could be made during the complex formation. Later, we transform the scores matrix into a fixed length input vector suitable for further statistical data analysis (aggregated values over columns, diagonals, etc.), and we identify the network properties (e.g., sizes of connected components) using interaction graph. This data is used by the level-II predictor, which integrates information similarly to a human expert. Figure 1 presents this pipeline graphically.

Figure 1 Schematic depiction of our two-stage ensemble method.

Recently, several machine learning algorithms were applied to predicting protein interactions. Our study takes similar route to Yip et al. (2009) who predicted interactions of yeast proteins at the level of residues, domains, and whole proteins. On the residue level sequence and secondary structure were used; on the protein level they used phylogenetic profiles, sub-cellular localisations, gene expression data and interactome-derived features; finally, on the domain level, the co-evolution was selected to characterize interacting proteins. They constructed a classifier allowing for the information flow between the above three levels to improve the final prediction. This approach was further developed by Saccà et al. (2014), who introduced a different model of knowledge integration, and demonstrated its superiority on the previously used benchmarking data. Reported AUC values reached 0.80 for residues, 0.96 for domains and 0.82 for proteins. However, these results are likely to be biased because in the testing phase the authors used different interactions but the same protein set as in the training phase. It is unknown how their method would perform on previously unseen proteins.

Our method differs from the previous approaches in several ways. First, we treat residue level predictions only as the input for identification of protein level interactions. The information flows only bottom-up: from the residue level to the protein level. In Yip et al. (2009) work information flow between levels was an additional technique to improve prediction, in our method it is the core of the prediction. Moreover, we use only sequence and secondary structure features for prediction; PDB data is only used in training, which makes it possible to apply our algorithm even if the three-dimensional structures of both proteins are unknown. The secondary structure can be predicted with high accuracy by the standard bioinformatics methods. Therefore, we can use our method even when the high level properties of proteins remain unknown.

We compare our results with PPI predictors targeted at yeast which exploit global sequence properties (i.e., not considering local residue interactions). One such methods was developed by Liu (2009). He constructed the feature vector by comparing values of the selected amino acid indices at selected distances in the sequence. The data contained 5926 interacting protein pairs from DIP database. His method achieved 0.87 precision and 0.90 recall. We reimplemented his algorithm for comparison purposes.

Another sequence-based method was proposed by Chang, Syu & Lin (2010). Their data contained 691 protein interactions. The authors used sequence composition and the accessible surface area predicted by another algorithm (Chang et al., 2008) as base features, which were averaged over the entire sequence. They observed 0.72 precision and 0.80 recall. Part of their work not relying accessible surface area was also reimplemented here.

Other popular protein sequence representations used for predicting protein–protein interactions include Pseudo Amino Acid Composition (Zhao, Ma & Yin, 2012), 2-grams (Nanni, Lumini & Brahnam, 2014) and Quasiresidue Couples (Guo & Lin, 2005). We implemented all of these techniques and evaluated their performance on our own datasets.

Many other algorithms solving the same problem, but operating on different principles, were developed. One of the possible examples is PIPE predictor Pitre et al. (2006), discovering protein interactions in yeast using large sequence motives database, achieving overall accuracy 75%. Ultimately, we are interested in comparing our approach with methods belonging to this class, however such comparison is not a straightforward procedure. This is because of the conceptual differences between our methods—for knowledge-based approaches it is impossible to separate the algorithm from the database it relies upon. In this paper we focused on comparing our predictor with methods based primarily on machine learning which are possible to evaluate naturally within our experimental framework.

Here we would like to stress one of the most important outcomes of our study. Namely, the performance results reported in various publications are generally not comparable directly. They tend to differ in collected data methodology, definition of positives and negatives and evaluation procedures. In our work, we focused first on preparing the collection of high quality interactions data and removing any bias from the further evaluation procedure. To obtain a meaningful comparison between different methods, we reimplemented several schemas for aggregating features of protein sequences (including the method of Liu (2009)), and evaluated them on our benchmarking dataset. Our method outperformed the others in terms of ROC AUC by a large margin (the smallest difference was 0.70 AUC for our method vs 0.60 AUC for Liu’s method).

The performance scores obtained in our study are much lower than usually reported in the literature. We claim that this is an effect of carefully balancing positives and negatives in our datasets and the rigorous evaluation strategy in which a predictor is always tested on proteins unseen during the training phase. What we are measuring is the ability to predict real protein compatibility and not just relative proteins’ reactivity. This task is much harder, and we demonstrate that popular methods using sequence-derived features do not perform well in this context. Our result confirms previous methodological studies in this area (Park & Marcotte, 2012).

Materials

We extracted three dimensional (3D) structures of all yeast protein hetero-complexes from Protein Data Bank (PDB) (Berman et al., 2000), which were crystalised using X-RAY with the resolution below 3 Å. Homologous structures were removed with 90% sequence identity threshold. We mapped the PDB complexes to Uniprot (The UniProt Consortium, 2014) ids using SIFTS mapping (Velankar et al., 2013) tool. Secondary structure was extracted from PDB structures using DSSP software (Joosten et al., 2011; Kabsch & Sander, 1983). Because of the gaps present in PDB structures gaps in the extracted secondary structure also occurred. Such gaps were filled with the coil symbol. For evaluation purposes we also employed PSIPRED (Jones, 1999) to predict secondary structure from protein sequence.

On the residue level, from 3D structures it was possible to extract all interacting pairs of residues. We considered any two residues from two distinct proteins as interacting if they were located in Euclidean space within a distance threshold of 4 Å from each other. On the protein level, we identified two proteins as an interacting pair if there was at least one residue level microscopic interaction between them. We were interested only in heterodimers, i.e., interactions occurring between two different proteins. While predicting homodimers is also valuable, we decided to leave out homodimers and focus on heterodimers for two reasons: (a) hetero-interactions can contribute a lot to our understanding and the reconstruction of the true protein interaction network (PIN), (b) in our data homodimers occurred much more frequently and there was a risk that they will dominate heterodimers completely.

Residue level positives and negatives

The first step of our procedure was to build a training dataset for level-I predictor. We employed the sliding window technique to extract fragments of protein sequence. In this work we refer to it as extraction window.

Positive examples in the training set were formed from pairs of fragments in which central residues interact, and additionally certain number of other residues within a specified distance from the central one interact. We refer to the required number of interacting residues as interaction threshold and to the maximal distance from the central residue as the maximal neighbour interaction distance. By introducing this restriction we deliberately focused on strong interactions, filtering out the weaker ones, which could be just noise in the crystallisation process.

In the data preparation phase, we fixed the maximal neighbour interaction distance at 10 residues. We have chosen this value following the studies of Youn et al. (2007) and Kauffman & Karypis (2010), who predicted binding residues from protein sequences and used the window of size 21 (1 central residue, 10 residues to the right, 10 to the left). As for interaction threshold, we tested the values of 0, 5, 10, 15, and 20 interacting residues within the maximal interaction distance.

Let us observe that the values of the maximal neighbour interaction distance and extraction window size are not necessary the same. One can imagine identifying residue level positives using larger interaction distance, and then encoding features of only a few central residues using small extraction window, and vice versa. Indeed, in our work we tested different sizes of extraction window ranging from 3 to 31 independent of the fixed interaction distance.

In our data extracted from PDB we had no natural source of negative residue interactions. Pairing sequence fragments fully randomly could result in a lot of noise and false negatives. Therefore, we decided to extract non-interacting pairs of fragments only from interacting protein pairs. Fragment pairs without any interacting residues or fragment pairs in which one fragment has some interactions but the other has none were considered negatives. This way it was guaranteed that at least one of the fragments did not come from the interface region. The number of potential negatives was much larger than the number of positives, therefore we decided to keep the imbalance ratio at 3:1. The required amount of negatives was therefore sampled at random. This is a common practice in machine learning since most of the algorithms perform poorly on datasets with large class imbalance (see Chawla (2005) for review).

Protein level positives and negatives

The second step of our methodology was preparing data for training level-II predictor. We used the same dataset of interacting protein pairs from PDB database as positives, and generated negative examples. The construction of high quality negative examples is very difficult. Common methods for generating negatives include drawing random pairs of biomolecules from all known proteins found in a specific organism (Saha et al., 2014), or only from the selected subset of the whole proteome, namely from the proteins occurring in positive examples (Chang, Syu & Lin, 2010). We strongly believe that such methods have their inherent drawbacks because they ignore network properties of the underlying protein interactome. We used the following procedure instead:

• Let G1 be a graph representing positive examples. Denote V = v1, …, vn as the set of its vertices. Each vertex in V represents a protein and each edge vi, vj represents an interaction. Let [Deg(v1), …, Deg(vn)] be a vector containing degrees of vertices from V. Let G2 be a graph of negative interactions. At first it has vertices identical to G1 and no edges.

• While there exist v such that Deg(v) > 0:

1. Find vertex v with the largest Deg(v).

2. Find vertex u if exist such that:

(a) There is no edge (v, u) in G1.

(b) u has as large Deg(v) as possible.

(c) Distance d(u, v) in G1 is as large as possible.

3. If u exist:

(a) Add edge (u, v) to G2.

(b) Deg(v)←Deg(v) − 1

(c) Deg(u)←Deg(u) − 1

4. else: Deg(v)←0

Such schema of constructing negative protein pairs set is unbiased, i.e., the protein composition of the positives and the negatives remains identical. Every single protein has the same number of positive and negative interactions. This forces the trained classifier to predict meaningful biophysical interactions rather than predicting general reactivity (the relative number of interactions) of a single protein. Otherwise, the best results would be achieved by a predictor, which predicts that the two proteins interact if each of them has a lot of interactions in general, regardless of their actual compatibility. It’s also important that our algorithm favours protein pairs which are remote to each other in the interaction network which reduces the risk of introducing false negatives.

Train-test split

The last step of data generation procedure is splitting samples between training and testing sets. In order to truly evaluate our method in a realistic setup, we split the benchmarking dataset at the protein level, not at the residue level. This made our goal more difficult as compared to previous works that often used residue-level splitting of benchmarking dataset. The schematic depiction of the train-test split is given by Fig. 2. The Level I classifier was evaluated through train-test experiment with a relatively large datasets. The Level-II classifier was evaluated through cross-validation experiment which used the testing set only—information from the training set came only in the form of the trained level-I classifier. Such schema eliminated the risk of overoptimistic performance estimates caused by the same data appearing during training and testing phases.

Figure 2 Schema of train-test split for evaluation of trained classifiers.

Numbers of examples used in each step are given in parentheses.

Methods

Level-I predictor was trained to recognise interacting pairs of fragments. This should be the equivalent of detecting compatible protein patches on the surface of a protein. During prediction, for each possible pair of fragments from two different proteins a prediction is made, and the likelihood estimation for all against all pairs of fragments are stored in the interaction matrix. The Level-II predictor uses the output of level-I predictor, predicting binary interactions between two proteins using the aggregated features, i.e., the complementarity between their surface patches.

Level-I features

We trained the level-I predictor on interacting sequence fragments of proteins from the training set and tested it on the testing set. Input for level-I predictor consisted of pairs of sequence fragments of the length of extraction window.

The following sets of features were considered:

• Raw sequence—raw sequence of amino acids encoded numerically.

• HQI8—sequence of amino acids encoded with High Quality Indices (Saha, Maulik & Plewczynski, 2012). These features were constructed by clustering AAindex database (Kawashima et al., 2007). Each amino acid is described by 8 values representing its physicochemical and biochemical properties.

• DSSP structure—secondary structure of the protein extracted from PDB complex with DSSP software. It was limited to the three basic symbols: E—β-sheet, H—α-helix, C—coil.

• PSIPRED structure—secondary structure of the protein predicted from sequence with PSIPRED software. It was limited to the three basic symbols: E—β-sheet, H—α-helix, C—coil.

As the core classifier, we evaluated two popular machine learning methods: Random Forest and Support Vector Machine. Both algorithms are commonly used in bioinformatics and are considered the best off-the-shelf classifiers (Yang et al., 2010). Their parameters values were chosen through a grid search. As the performance measure we have chosen ROC AUC (area under Receiver Operating Characteristic curve).

Level-II features

To infer a binary interaction between two proteins, we consider all possible interactions between their sequence segments as predicted by level-I predictor. An output of this phase is a matrix of likelihoods with the dimension equal to the multiplied proteins’ lengths. Each prediction score is a real number between 0 and 1. Sample matrices of scores for a positive and negative case are presented in Fig. 3.

Figure 3 The level-I prediction matrices for two protein pairs.

White colour corresponds to score 0.0, black colour corresponds to score 1.0.

To transform the 2D matrix into an input vector suitable for level-II predictor, we extracted the following features (numbers in parentheses denote the number of values in the final feature vector):

• the mean and variance of values over the matrix (2),

• the sums of values in 10 best rows and 10 best columns (20),

• the sums of values in 5 best diagonals of the original and the transposed matrix (10),

• the sum of values on intersections of 10 best rows and 10 best columns (1),

• the histogram of scores distributed over 10 bins (10),

• graph features: fraction of nodes in the 3 largest connected components (3).

The graph features require further explanation. Predicted contacts between residues were represented as a bipartite graph. Nodes in the graph represented residues and edges represented predicted contact. To make the graph more realistic biologically, for each node we left only 3 strongest outgoing edges. We set the value of this threshold (3) following the observation that in our PDB structures the mean number of interactions of a single interacting residue is between 2 and 3. In the trimmed graph we calculated fractions of nodes contained in 3 largest connected components. Those values were also appended to the feature vector.

Evaluation of the level-II predictor

The performance of the level-II predictor was evaluated through a variant of stratified 30-fold cross-validation performed on the protein level. Each fold contained 130 of positive protein pairs and 130 of negative protein pairs from the testing set. There was no overlap between splits on the pair level, but there was still an overlap on the level of single proteins which constitute pairs. We observed that this introduced a huge bias into evaluation results (similar observation was previously made by Park & Marcotte (2012)). To fix our cross-validation scheme we applied the following procedure:

1. Let O=p11,p12,p21,p22,…,pn1,pn2 be a set of all protein pairs.

2. For each fold F ⊂ O:

(a) Build a set P composed of all proteins occurring in F: P=x:∃yx,y∈F∨y,x∈F.

(b) Build a set A ⊂ O composed only of pairs consisted of proteins occurring in P: A=x,y:x∈P∧y∈P.

(c) Build a set B ⊂ O composed only of pairs consisted of protein not occurring in F but occurring in the testing set: B=x,y:x∉P∧y∉P∧x,y∈O.

(d) Train the classifier on B set, and test it on A.

3. Collect all the predictions for A-sets, and calculate performance metrics.

The above described procedure differs from the standard cross-validation, since the number of observations in constructed test sets vary slightly, but this variance is small, and does not influence the estimated performance. Such evaluation schema does not allow for any information leak: the datasets are always balanced, and the classifier is tested on previously unseen proteins.

As classification methods for level-II predictor we used Random Forest and Support Vector Machine with parameters tuned through a grid search.

Protein sequence feature aggregation

We compared our ensemble method with various sequence feature aggregation schemas that are commonly applied in machine learning predictors of proteins interactions. To make the benchmarking results comparable between different algorithms, we used the same classification method (Random Forest) and evaluation procedure (modified 30-fold cross-validation on the testing set) as for level-II predictor. We benchmarked the following feature aggregation schemas:

1. AAC—Amino Acid Composition (Nanni, Lumini & Brahnam, 2014). Feature set is the set of frequencies of all amino acids in the sequence.

2. PseAAC—Pseudo Amino Acid Composition (Chou, 2001). Feature set consists of the standard AAC features with k-th tier correlation factors added. We calculate those correlations on HQI8 indices.

3. 2-grams (Nanni, Lumini & Brahnam, 2014). Feature set comprises of frequencies of all 400 ordered pairs of amino acids in the sequence.

4. QRC—Quasiresidue Couples (Guo & Lin, 2005). A set of AAIndices is chosen. For each index d combined values of this property d for a given amino acid pair are summed up for all the pair’s occurrences over the full protein sequence. Occurrences for pairs of residues separated from each other by 0, 1, 2…m residues. In effect, one obtains QRCd vectors of length 400 × m. In this model we also use HQI8 indices.

5. Variation of Liu’s protein pair features (Liu, 2009). The method starts from encoding each amino acid in a protein sequence with 7 chosen physicochemical properties, thus obtaining 7 feature vectors for each sequence. For each feature vector its “deviation” is calculated: γdj=1n−d∑i=1n−dxij×xd+i,jj=1,…,7d=1,…,L

where xij is the value of descriptor j for amino acid at position i in sequence P, n is the length of protein sequence P, and d is the distance between residues in the sequence. For the purpose of the comparison, we tested this method with the original 7 amino acid indices used by Liu, as well as with HQI8 features. We tested different values of L from 5 to 30 in a quick cross-validation experiment on our data and chose L = 9 as yielding the best results.

Results and Discussion

Experimental results

We evaluated carefully all subsequent steps of our method to choose optimal features and parameter values. Then we compared performance of level-II predictor with popular sequence encoding schemas. In our experiments we stuck to the rule that during classifier training we can use all the information available in PDB complexes, but in the evaluation phase only information derived from the sequence is allowed. This was to demonstrate that our method can be employed successfully in a situation when only protein sequences are known.

The first task was to decide on the set of optimal features for the level-I predictor. We have chosen ROC AUC as the performance metric. To have a complete picture, we measured both the performance of level-I predictor and the performance of level-II based on the trained level-I predictor. We tested two sources of secondary structure information—DSSP and PSIPRED—separately, but the evaluation was done using PSIPRED secondary structure (results on DSSP are given in parentheses for completeness). Initial experimentation was done with Random Forest, then we tested whether it is possible to gain anything by replacing it with a carefully tuned SVM.

Table 1 ROC AUC scores of level-I predictor trained on different sets of features.

Interaction threshold was set to 15. Extraction window size was set to 21. A+B denotes feature vector constructed by concatenating two sets of features. RF—Random Forest, 300 trees, maximum tree depth 15, SVM—Support Vector Machine, RBF kernel, C = 2, γ = 0.048. For level-II Random Forest with 300 trees and maximum tree depth 7 was used. Main scores were calculated for PSIPRED-predicted secondary structure, values in parentheses concern scores for DSSP secondary structure.

Classifier	Features	Lvl-I AUC	Lvl-II AUC	
RF	Raw sequence	0.64	0.59	
HQI8	0.70	0.59	
PSIPRED structure	0.67	0.63	
PSIPRED structure + Sequence	0.69	0.60	
PSIPRED structure + HQI8	0.72	0.56	
DSSP structure	0.72 (0.87)	0.70	
DSSP structure + Sequence	0.73 (0.87)	0.65	
DSSP structure + HQI8	0.74 (0.85)	0.64	
SVM	DSSP structure	0.59 (0.84)	0.57	

Evaluation results of the predictor on different features for 5 interaction thresholds are presented in Table 1. The DSSP secondary structure performed better than sequence features. Including both secondary structure and HQI8 in the feature vector provided minor improvement on level-I, but it resulted in performance degradation on level-II. While the quality of level-II prediction is based on the quality of level-I prediction, our results show that overfocus on optimizing level-I prediction may result in overfitting visible on level-II. SVM did not perform better than Random Forest. In all further experiments we used secondary structure as the only source of features for level-I predictor, together with Random Forest as the learning algorithm.

Table 2 ROC AUC scores of level-I predictor for different interaction thresholds.

DSSP-extracted secondary structure was used for constructing feature vector. Extraction window size was set to 21. For level-I Random Forest, 300 trees, maximum tree depth 7 was used. For level-II Random Forest with 300 trees, maximum tree depth 7 was used. Main scores were calculated for PSIPRED-predicted secondary structure, values in parentheses concern scores for DSSP secondary structure.

Threshold	Lvl-I AUC	Lvl-II AUC	
0	0.67 (0.84)	0.67	
5	0.67 (0.85)	0.67	
10	0.69 (0.86)	0.68	
15	0.72 (0.87)	0.70	
20	0.75 (0.88)	0.64	

Table 2 reports ROC AUC values for different interaction thresholds. Each value changes the definition of positive and negative example for level-I predictor. The larger the threshold, the more positive and negative examples differ. We can see that increasing interaction threshold indeed made level-I prediction task easier, but the impact on level-II prediction was non-linear. We chose 15 as the optimal threshold value.

Figure 4 ROC AUC scores of level-I predictor trained on secondary structure for different extraction window sizes.

Random Forest was used as the classifier.

The final decision was selecting the optimal extraction window size. Figure 4 presents level-I ROC AUC score against window size. After analysing the plot we decided to keep size 21 as it provided good performance and it was previously used in other publications. In that way we fixed extraction window size to the same value as the maximal interaction distance, used to define positive residue interactions.

After fixing the parameters, we wanted to choose classification algorithm for level-II predictor and compare performance of our multi-level representation with representations based on the aggregated protein sequence. Results are presented in Table 3. Once again Random Forest proved to be more suitable for the task than Support Vector Machine. On this kind of data level-II predictor outperformed other methods significantly.

Table 3 Performance scores.

t = x denotes interaction threshold of x interacting residues. Level-II predictor used secondary structure predicted by PSIPRED. RF—Random Forest, 300 trees, maximum tree depth 7, SVM—Support Vector Machine, RBF kernel, C = 1, γ = 2.

Clf	Features	Accuracy	Precision	Recall	AUC	
SVM	Lvl-II pred (t = 15)	0.55	0.58	0.55	0.57	
AAC	0.54	0.56	0.66	0.54	
PseAAC	0.54	0.55	0.61	0.55	
2grams	0.55	0.56	0.64	0.55	
QRC	0.51	0.53	0.59	0.53	
Liu’s dev (HQI8)	0.55	0.57	0.60	0.56	
Liu’s dev (original)	0.55	0.57	0.60	0.56	
RF	Lvl-II pred (t = +15)	0.68	0.70	0.68	0.70	
AAC	0.54	0.57	0.54	0.56	
PseAAC	0.53	0.55	0.52	0.55	
2grams	0.53	0.56	0.49	0.55	
QRC	0.50	0.52	0.43	0.51	
Liu’s dev (HQI8)	0.55	0.58	0.55	0.60	
Liu’s dev (original)	0.56	0.59	0.57	0.59	

Role of secondary structure

Our results stress the link between secondary structure and residue contacts. Compatibility of structural motives is important for contact forming. To get more insights from our results, we extracted feature importances from the trained Random Forest model of level-I predictor. Relative importance of an attribute in a decision tree is expected fraction of examples split by nodes based on this attribute. In Random Forest this value is averaged over all trees. Relative feature importances for the input vector constituted by secondary structure annotation are given by Fig. 5. It is visible that the most important are central and border residues of the fragments.

Figure 5 Relative importances of individual features in level-I predictor feature vector.

To understand what is happening on these positions, we calculated frequency of particular secondary structure patterns occurring between the residues of interacting and non-interacting fragments. Calculated frequencies are presented in Table 4. It is clear that pairs H–H occur more frequently among interacting fragments than non-interacting for all three positions. E–E pair is more likely to occur in interacting fragments on the central position, but not so much on the border. C–C pair is generally more prevalent among non-interacting fragments, but this effect is also stronger for central residues. Those results are not surprising: since secondary structure motives are stabilised by hydrogen bonds the relation to both intra-molecular and inter-molecular contacts is expected (Hubbard & Kamran Haider, 2010).

Table 4 Secondary structure patterns for interacting and non-interacting fragments.

ai denotes structural motive of the i-th residue of one fragment, bj denotes structural motive of the i-th residue of the other fragment.

Pattern	Interacting	Non-interacting	
a0 = H∧b0 = H	0.13	0.09	
a0 = H∧b0 = E	0.04	0.05	
a0 = H∧b0 = C	0.16	0.17	
a0 = E∧b0 = E	0.06	0.03	
a0 = E∧b0 = C	0.08	0.09	
a0 = C∧b0 = C	0.23	0.30	
a10 = H∧b10 = H	0.17	0.08	
a10 = H∧b10 = E	0.03	0.04	
a10 = H∧b10 = C	0.13	0.17	
a10 = E∧b10 = E	0.14	0.03	
a10 = E∧b10 = C	0.09	0.09	
a10 = C∧b10 = C	0.18	0.31	
a20 = H∧b20 = H	0.15	0.08	
a20 = H∧b20 = E	0.06	0.05	
a20 = H∧b20 = C	0.18	0.18	
a20 = E∧b20 = E	0.05	0.03	
a20 = E∧b20 = C	0.08	0.10	
a20 = C∧b20 = C	0.21	0.31	

From our experiments it is clear that using the real secondary structure for classifier training is preferable over the predicted secondary structure, even if later only predicted secondary structure will be available. This limits the noise introduced during the training phase allowing the predictor to focus on the important patterns.

What is surprising at first glance is that encoding both secondary structure and protein sequence in a single feature vector improves the prediction on level-I but leads to a worse prediction quality on level-II. It is true that protein sequence contains more information than just secondary structure, but, on the other hand, including more attributes increases the risk of overfitting. This is the case of so-called bias–variance trade-off, where increasing the model complexity potentially decreases classifier bias but at the same time increases classifier variance (Geman, Bienenstock & Doursat, 1992). Here, the situation is even more complex because overfitting may occur at different levels—improving the prediction quality at level-I may in some cases make level-II task more difficult. From various feature combinations considered in our study secondary structure alone proved to be optimal, conveying important information while keeping the model complexity low.

We may speculate that this phenomena comes directly from the hierarchical nature of biological systems. Biological processes have to be robust and predictable, harnessing the dynamic of physical molecules in a constructive way. It is easier to build the upper layers of a complex system using constrained and standardized building blocks. Secondary structure motives are such stable building blocks, more constrained than the protein sequence. In that sense secondary structure may be seen as an useful data compression optimized evolutionary to perform certain functions. Our classifier performs better with a more compact, specialized representation than with the full sequence information, which requires too many free parameters to estimate.

Quality of residue contacts prediction

In our work we focused on predicting binary protein interaction and the constructed residue-level predictor served only to generate data for level-II predictor. We optimised level-II prediction quality and demonstrated that optimising level-I only could lead to overfitting on level-II. In fact, for the threshold values greater than 0, our level-I predictor was not trained to identify single residue contacts anymore; it focused on discovering only strongly interacting fragments important for inter-protein interactions. To assess the importance of this difference in task formulation, we calculated ROC AUC score for level-I predictor not on our filtered segment pairs data but on real contact matrices of the test proteins. The predictor obtained 0.65 AUC—a score lower than on the filtered data and lower than level-II predictor score. This means that level-I predictor produces useful information for predicting protein–protein interactions but is not necessarily a good predictor of specific residue contacts.

After examining predicted contact maps visually, we realised that they consist of characteristic horizontal and vertical lines. This means that certain sequence fragments are predicted to be very active and likely to interact while the others are classified as inactive. We considered the possibility that our predictor focused on predicting solvent exposure of the fragments. To verify this hypothesis we calculated relative solvent accessibility for each residue (Shrake & Rupley, 1973). We chose threshold value 0.2 to distinguish between buried and accessible residues (Chen & Zhou, 2005). All residues of proteins from our test set were divided into these two classes. Then we tested whether mean values of rows and columns of the predicted contact matrices are useful for predicting residue state (buried or accessible). Surprisingly, we obtained AUC ROC score 0.46, which means that characteristic rows and columns in our matrices are not directly correlated with just solvent accessibility.

It is possible that our level-I predictor reflects more specific protein properties such as the location of active sites. Improving our method would require an in-depth analysis of these matters. Since solvent accessibility is not correlated with the current predictions perhaps it is possible to include this information to further improve accuracy.

Quality of protein interactions prediction

We draw the reader’s attention to the fact that the performance of popular protein representation strategies evaluated on our data was generally much lower than results reported in the literature. One of the reasons may be a relatively small size of our dataset—it might not contain enough examples for a classifier to learn complex patterns. The other explanation is the way we constructed positives and negatives. In our case every protein occurred in the same number of positive and negative pairs. Moreover, we performed cross-validation with splits on the protein level, which means that no single protein occurred simultaneously in training and testing set. In such conditions any method which makes a good prediction of general proteins’ reactivity but does not consider their actual compatibility performs poorly. This observation is consistent with results obtained by Park & Marcotte (2012), who analysed the impact of performing splits on the component level instead of on the pair level on cross-validation results. They reported AUC scores of popular protein interaction prediction methods dropping from 0.7–0.8 to 0.5–0.6. We strongly believe that performing splits on the component level, i.e., keeping training and testing protein sets disjoint, results in a better estimation of predictor performance in a realistic scenario of predicting interaction between two previously unseen proteins.

Even though our evaluation procedure was carefully designed to reduce certain biases, we still do not expect it to reflect the reality of protein–protein interaction prediction perfectly. The most important issue is the proportion of positives to negatives in our datasets. On the protein level, we used balanced sets with the same number of positives and negatives. In reality the number of negatives, i.e., pairs of non-interacting proteins is much larger than the number of positives. Grigoriev (2003) estimated that there are around 26,000 hetero interactions between 6,300 proteins in yeast proteome. This would make the ratio of negatives to positives equal to: 6,3002−26,00026,000=762.1481.

It is not feasible to construct a dataset with such a ratio in practice. Since the negatives in our experiment are constructed through pairing proteins randomly, there might be false negatives present. The more negatives we draw, the more false negatives we will introduce. In the most pathological case all the unknown interactions we would like to predict will be labelled as negatives. One of the possible solutions to this problem is to calculate performance metrics using balanced data and then introduce corrections based on the expected negative to positive ratio.

Classifier performance is calculated from a confusion matrix consisting of the following terms: TP—true positives (positives classified as positives), TN—true negatives (negatives classified as negatives), FP—false positives (negatives classified as positives), FN—false negatives (positives classified as negatives). The correction involves multiplying terms based on negative examples in the data set—TN and FP—by constant d = 762. Let us examine the definitions of the specific performance metrics: Accuracy=TP+dTNTP+dTN+dFP+FN

Recall=TPTP+FN

Precision=TPTP+dFP.

AUC is the area under receiver operating characteristic, which is a plot of recall (sensitivity) vs 1-specificity, where specificity is defined as: Specificity=dTNdTN+dFP=TNTN+FP.

From this equation it is clear that class imbalance does not affect recall and AUC. In the case of the final level-II predictor TP = 179, FP = 78, TN = 160, FN = 83, the corrected precision is 0.003, the corrected accuracy is 0.67. Precision score 0.003 is hardly impressive but it is still more than 2 times bigger than the score 0.0013 expected by chance when labelling protein pairs blindly. This means that the use of our predictor introduces valuable information.

In the light of these results, we believe that current methods for predicting protein–protein interactions from sequences, including ours, are not mature enough for large-scale application. The goal of reconstructing full protein interaction network is right now beyond reach. However, the developed predictors could still be applied in a variety of contexts. One of the possible applications is the initial screening of candidates for more costly in vitro experiments—choosing protein pairs for experimental verification in a more informed way would make the procedure more effective. Another possibility is using the predictor as a filter to improve quality of the interaction data collected from the existing databases. This is also an important issue because the quality of results from high-throughput experiments is still limited.

Conclusions

In this work we presented a method for constructing a multi-level classifier for protein–protein interactions. We demonstrated that the information present at the lower level can be successfully propagated to the upper level to make reasonable predictions. No additional features other than protein secondary structure predicted from sequence were required.

Our goal, predicting actual compatibility between two proteins regardless of their relative reactivity, forced us to collect high quality data and develop a rigorous evaluation procedure. We have taken into account properties of protein interaction network to construct balanced negatives. During the evaluation we carefully separated training and testing proteins to avoid information leak. We demonstrated that our method is working under such conditions better than popular sequence feature aggregation schemas.

There is still much room for further improvements regarding classification accuracy. We plan to include additional features both at the residue level and at the protein level to see if our model can benefit from them. Another direction that we want to explore is expanding the model to include proteins from organism other than yeast, and evaluating it on bigger datasets.

We hope that our work will inspire further discussion regarding evaluation strategies for protein interaction predictors. We believe that deeper understanding of those matters would allow the comparison of different methods in a more systematic manner which would be beneficial for the research done in this area.

Additional Information and Declarations

Competing Interests

Author Contributions

Data Deposition

The authors declare there are no competing interests.

Julian Zubek and Marcin Tatjewski performed the experiments, analyzed the data, wrote the paper, prepared figures and/or tables, reviewed drafts of the paper.

Adam Boniecki and Maciej Mnich analyzed the data, contributed reagents/materials/analysis tools.

Subhadip Basu conceived and designed the experiments, wrote the paper.

Dariusz Plewczynski conceived and designed the experiments, contributed reagents/materials/analysis tools, wrote the paper, reviewed drafts of the paper.

The following information was supplied regarding the deposition of related data:

Github: http://zubekj.github.io/mlppi/.

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
