# Peer review of "Multi-level machine learning prediction of protein–protein interactions in Saccharomyces cerevisiae"

_PeerJ, doi:10.7717/peerj.1041_

## Round 0.1 · original submission · Major Revisions

Both reviewers comment on the surprising result that secondary structure alone seems to be sufficient to generate the best predictions. It will be important to really get to the bottom of this issue. What is going on there?

Reviewer 1 ·

Basic reporting

The authors present two algorithms for predicting fragment-fragment and protein-protein interactions based on information derived from sequence alone. The predictive performance of the two algorithms is rather modest (AUC 0.70 for protein-protein interaction prediction, AUC 0.88 for fragment-fragment interaction prediction). Surprisingly, only secondary structure information is needed for optimal prediction, not sequence information, and it is not clear why this is the case.

There is extensive literature on sequence-based prediction of protein-protein interaction that the authors did not cover. For example, the PIPE algorithm (Pitre et al., BMC Bioinformatics 2006) uses recurring sequence fragments between known interacting protein pairs to predict new protein-protein interactions.

The Results and Discussion section is short. This section should be expanded to thoroughly and rigorously discuss how the conclusions are supported by the data, possible alternative hypotheses, the advantages and disadvantages of the proposed algorithms, and broader implications for practical prediction of protein-protein interactions and protein-protein interaction sites.

There are quite a few spelling, grammatical, and formatting errors throughout the text and references. For example:
* Page 2, Introduction section: "Yeast" in italic should be "yeast".
* Page 4, "Residue level positives and negatives" section: ".. could result in lot noise and false negatives" should be ".. could result in a lot of noise and false negatives".
* Page 7, "Level-II predictor" section: "Level-II predictor" actually describes the content of the previous section.
* Page 10, Acknowledgments section: "This paper is founded by" should be "This paper is funded by".
* Page 10, Acknowledgments section: "and and" should be "and".
* Page 10, References section: The Guo & Lin reference is missing the journal name.

Experimental design

In the Abstract, the authors state that "prepared datasets and source code for our experimental pipeline are freely available for download from URL provided from authors upon request (open source Python implementation, OS independent)". The authors should directly provide the URL in the manuscript. Alternatively, key datasets and/or source code should be submitted to the journal as supplementary materials.

Validity of the findings

The predictive performance of the algorithm for protein-protein interaction is rather modest (AUC 0.70). This is not going to be very useful as a practical tool for predicting protein-protein interactions. In practice, the ratio of interacting protein pairs to non-interacting protein pairs is estimated to be on the order of 1:100 to 1:1000. This means that only a very small percentage of protein pairs are interacting, and the vast majority of protein pairs are not interacting. Even a small false positive rate will give rise to a huge number of false positives which will overwhelm true positive predictions. This practical aspect of controlling false positive rates for protein-protein interaction prediction is not discussed in the paper.

The predictive performance of the algorithm for fragment-fragment interaction is more promising and potentially more useful in practice (AUC 0.80). However, there are a number of alternative hypotheses that the authors should consider. For example, it is possible that the authors are simply predicting the solvent exposure of these fragments. And since only exposed fragments can interact with each other, this information alone will give rise to a decent predictive power.

A surprising result is that secondary structure alone is sufficient to generate the best predictions, and the sequence information is not needed. But it is unclear why this is the case. Is it possible that the algorithm is simply predicting loop residues? And if interacting sites are enriched for loop residues, then this information alone will give rise to a decent predictive power.

In Table 2, the predictive performance of the proposed algorithms are similar to other previously published algorithms when SVM is used, but are better when Random Forest is used. In machine learning, it is not recommended to use the test set to choose the best-performing classifier and then report the performance on the test set only. Rather, the performance of the chosen optimal classifier should be evaluated independently using a third, validation set. This is especially important when different classifiers give very different results which can potentially alter the major conclusions, as is the case here.

·

Basic reporting

The manuscript is well written and generally clear. However, it would benefit from a closer checking of grammar and usage throughout. For example, articles are frequently omitted or misused as in the following: "They constructed classifier allowing for the…", where there should be an "a" following "constructed".

Experimental design

The manuscript describes an important problem, that of predicting interactions between proteins, and uses a novel two-level machine learning approach to do so. The results are interesting but the manuscript would be improved by addressing the following comments:
1. The authors use an artificially imbalanced training set and this balance (3:1, negatives:positives) and this balance is apparently maintained in the testing set and to generate the final performance values reported. This is not necessarily a huge problem, and, as the authors note, is common practice in the community. However, the authors should note what effects this will have on their performance metrics; it will not affect AUC but will cause accuracy to be artificially underestimated but can cause precision to be dramatically overestimated (see my post on this subject at http://jasonya.com/wp/another-word-about-balance/). These effects should be noted in the text and performance should be reported on the true balance of negative to positive examples to fully inform the readers.
2. For the level-I predictor a better description of the feature encoding is needed- how many features for each encoding, what were these features, etc.
3. The authors re-implemented several algorithms to allow for fair comparison between those algorithms and the current algorithm (first paragraph of the Protein sequence feature aggregation section). This seems like a very reasonable thing to do, but it should also be emphasized in the introduction- for example, when the previously reported AUCs for the existing methods are listed.
4. The authors mention that homodimers are excluded from their interaction sets. Is there are reason for this? These are also important interactions.
5. For the level-I analysis the authors do not present analysis of which features are most important for prediction beyond the individual encodings. This information would be very helpful for interpretation of the results and potentially would have broader implications (see my point on the validity of the findings below though).

Validity of the findings

My primary concern with the findings of this paper is that it seems that the best level-I predictor is based solely on secondary structure. This seems like a surprising and somewhat suspicious result. It is not clear to me how secondary structure alone would be predictive of protein-protein interactions. If this were true I would expect that it would have to be encoded in a pattern of secondary structure elements in some way, and those patterns would be very biologically interesting. This result/conclusion requires more evidence, analysis, and discussion in the test.

---

## Round 0.2 · accepted · Accept

Thank you for your careful revisions. Note that for the revised manuscript, both reviewers point out a few minor grammatical issues that I trust you will correct before final publication.

Reviewer 1 ·

Basic reporting

No Comments

Experimental design

No Comments

Validity of the findings

No Comments

Additional comments

My concerns have been addressed. I have one remaining minor comment:

On Page 11, Section "Role of Secondary Structure", the following sentence is not clear and needs to be revised: "Those results are not surprising: since secondary structure motives are stabilised by hydrogen bonds they relation to both intra-molecular and inter-molecular contacts is expected."

·

Basic reporting

Minor point: the authors frequently misuse "motive" instead of "motif" in the revised text. This should be corrected.

Experimental design

The revisions have adequately addressed my previous concerns.

Validity of the findings

The revisions have adequately addressed my previous concerns.